# Feasibility and Effect of Electroacupuncture on Cognitive Function Domains in Patients with Mild Cognitive Impairment: A Pilot Exploratory Randomized Controlled Trial

**DOI:** 10.3390/brainsci11060756

**Published:** 2021-06-07

**Authors:** Yujin Choi, In-Chul Jung, Ae-Ran Kim, Hyo-Ju Park, Ojin Kwon, Jun-Hwan Lee, Joo-Hee Kim

**Affiliations:** 1Clinical Medicine Division, Korea Institute of Oriental Medicine, Daejeon 34054, Korea; choiyujin@kiom.re.kr (Y.C.); arkim@kiom.re.kr (A.-R.K.); mable@kiom.re.kr (H.-J.P.); cheda1334@kiom.re.kr (O.K.); omdjun@kiom.re.kr (J.-H.L.); 2Department of Neuropsychiatry, College of Korean Medicine, Daejeon University, Daejeon 34520, Korea; npjeong@dju.kr; 3Korean Medicine Life Science, University of Science & Technology (UST), Campus of Korean Institute of Oriental Medicine, Daejeon 34054, Korea; 4Department of Acupuncture and Moxibustion Medicine, College of Korean Medicine, Sangji University, Wonju-si 26339, Korea; 5Research Institute of Korean Medicine, Sangji University, Wonju-si 26339, Korea

**Keywords:** mild cognitive impairment, electroacupuncture, acupuncture, non-pharmacological intervention, visuospatial function, randomized controlled trial

## Abstract

Although Electroacupuncture (EA) has been reported to be potentially effective for cognitive disorders, there is limited information about which domains of cognitive function can be improved by EA treatment. Sixty patients with MCI were randomly assigned (1:1:1) to groups to receive 24 sessions over 12 weeks of EA, sham EA, or usual care. In the EA group, electric stimulation was applied at bilateral PC6 and HT7. Various cognitive tests included in the Seoul Neuropsychological Screening Battery II (SNSB-II) were performed at baseline and post-treatment to explore effects of EA on five cognitive domains: attention, language, visuospatial function, memory, and frontal/executive function. Among 60 randomized participants (63.7 ± 7.1 years, 89.7% females), 45 (75%) completed the study. Of the five cognitive function domains of SNSB-II, the T score of visuospatial function showed a tendency to be higher in the EA group than in the usual care group at post-treatment assessment (mean difference: 10.16 (95% CI, 1.14, 19.18), Cohen’s *d* = 0.72, *p* = 0.0283). According to the results of this pilot study, the estimated effect size of EA on the visuospatial function of MCI patients compared to usual care was medium. Large-scale clinical trials are needed to confirm effects of EA on cognitive functions.

## 1. Introduction

Electroacupuncture (EA) is defined as the application of electric stimulation through acupuncture needles on acupoints. In general, EA has been thought to produce a higher intensity of stimulation on acupoints than manual acupuncture [1]. EA is commonly used for relieving pain [2]. Through neuroprotective mechanisms, acupuncture-related treatment has the potential to improve cognitive functions in animal studies [3]. Systematic reviews of randomized controlled trials have shown controversial but potential EA effects for improving the cognitive function of patients with mild cognitive impairment (MCI) and Alzheimer’s disease (AD) compared to anti-dementia drugs including nimodipine and donepezil [4,5,6,7].

MCI is a preclinical state of early dementia, which is defined by a noticeable decline in cognitive function considering age-related cognitive decline but maintaining independence related to daily living activities [8,9,10]. Early detection and intervention of MCI are clinically important to decrease the conversion rate to dementia. Cholinesterase inhibitors have been suggested for treating MCI. However, a recent systematic review concluded that cholinesterase inhibitors cannot produce a constant effect for patients with MCI [11]. There is a strong need for a new strategy to manage MCI and prevent its progression.

Cognitive function decline can occur in various domains. Changes in the memory domain might occur, leading to difficulties in tracking dates and appointments. Changes in the language domain might lead to difficulties in finding appropriate words or fluency. Changes in visuospatial function may lead to difficulties in drawing or identifying familiar persons. Changes in executive function may lead to difficulties in solving complex problems [12]. Amnestic MCI (aMCI) and non-amnestic MCI (naMCI) as clinical subtypes of MCI can be distinguished according to the presence of memory impairment. MCI patients who are having impairments in multiple cognitive domains can be distinguished from patients who are having impairment in a single cognitive domain [10,13]. aMCI has been reported to have a high risk of progression to Alzheimer’s disease (AD). Memory impairment is one of the main symptoms of AD [14]. Moreover, MCI patients who are having impairments in both memory and frontal/executive function have been reported to have a high risk of dementia conversion from MCI [15].

Previous research studies [4,5] on EA for MCI have mainly performed brief cognitive tests, including the Mini-Mental State Examination (MMSE) [16] and Montreal Cognitive Assessment (MoCA) [17]. MMSE and MoCA were developed to be brief screening tools for cognitive impairment. They are not complete neuropsychological examinations [18]. Thus, they cannot generate enough information for specific cognitive functions. A few studies have used the Clock Drawing Test (CDT), picture recognition [19], and the Alzheimer’s disease assessment scale cognitive subscale (ADAS-cog) [20]. However, they could not comprehensively evaluate various cognitive functions. Therefore, there have been limited reports about which cognitive function domains are improved by EA in patients with MCI. In addition, most previous studies mainly used active controls such as donepezil and nimodipine [4,5,19]. Few studies have used a sham control [21,22]. As a result, there is limited research about the specific effect of EA on cognitive impairment. Furthermore, most studies have a treatment period of 4–8 weeks [4,5,20]. Few studies have a long-term follow-up of more than 24 weeks.

In this pilot randomized controlled study, we explored the effect of EA on cognitive function, mainly focusing on each cognitive domain using a comprehensive neuropsychological examination. As a control, a sham EA group was used in addition to the usual care group to observe specific effects of EA. We carried out a 12-week treatment of EA with a follow-up of 24 weeks. The aim of this study was to evaluate the feasibility of a larger scale of clinical research and provide basic information on the efficacy and safety of EA treatment for patients with MCI.

## 2. Materials and Methods

### 2.1. Study Design

This study was a pilot randomized controlled trial with three parallel groups: an EA group, a sham EA group, and a usual care group. Using a protocol approved by the Institutional Review Board (Approval Number: djomc-142-1 and DJDSKH-16-BM-10) of Daejeon Korean Medical Hospital of Daejeon University, this trial was conducted in two centers. This trial was registered with the clinical research information service with a registration number of KCT0002164. We submitted our trial protocol to the registry site on 28 October 2016. After the review process, it was registered on 8 December 2016. The date of the first participant recruitment was 31 October 2016, which was after the submission date of the protocol.

### 2.2. Participants

Sixty eligible participants in two Korean medicine university hospitals from 31 October 2016 to 24 August 2018 were recruited. Inclusion criteria were: patients aged from equal to or more than 45 years old to under 80 years old; those who met the Peterson diagnostic criteria of MCI with memory problems for at least three months; Clinical Dementia Rating (CDR) score of 0.5 and Global Deterioration Scale (GDS) score of 2–3; Hachinski ischemic score ≤4; at least six years of education; and agreed with written informed consent. Diagnostic criteria of mild cognitive impairment included self-reported cognitive complaint, objective cognitive impairment, preserved independence in functional abilities, and no dementia [10]. In our pilot trial, objective evidence of cognitive impairment was judged by a MoCA score below 23 [23]. Participants with MoCA scores <23 were included. The following participants were excluded: (1) those with dementia by the Diagnostic and Statistical Manual of Mental Disorders-IV (DSM-IV); (2) those with a history of cognitive impairment due to any other cause (head trauma or brain injury, etc.); (3) those with hospital anxiety and depression scale-depression (HADS-D) ≥11; (4) those with a history of cerebral hemorrhage or infarction; (5) those with brain disorders including Parkinson’s disease, Huntington’s Disease, normal pressure hydrocephalus, brain tumor, and so on; (6) those with psychiatric illnesses such as major depressive disorder, schizophrenia, delusional disorder, and bipolar disorder; (7) those with a severe medical disease (diabetic complications; cardiovascular, hepatic, or renal disorder); (8) those with anemia, hypothyroidism, vitamin deficiencies, or malignancy; (9) those with any history of drug or alcohol dependence during the past 6 months; (10) those who received any traditional Korean medical treatment for MCI during the past 4 weeks; (11) those who were illiterate; (12) those who were involved in other clinical trials within 4 weeks; (13) those with inappropriate prosthesis for electro-acupuncture (pace-maker, a heart-lung machine, an electrocardiograph, etc.) or the possibility of a hypersensitivity reaction for electro-acupuncture including epilepsy; (14) pregnant, lactating women or those suspected to be pregnant; (15) those who had difficulties to comply with treatment, visits, or questionnaires.

### 2.3. Randomization and Allocation Concealment

Eligible participants were randomly allocated to the EA group, the sham EA group, or the usual care group in a 1:1:1 ratio. The randomization list was generated with a block randomization method using SAS Version 9.4 (SAS institute. Inc., Cary, NC, USA) by an independent statistician. After the randomization list was created, it was coded, sealed in an opaque double envelope and numbered in order according to the randomization list, and stored in a locked cabinet. Upon participant enrollment, an individual’s envelope was opened to determine his or her group allocation. The independent investigator who was in charge of the intervention verified group allocations of participants.

### 2.4. Blinding

Because of the characteristic of this study, clinical investigators could not be blinded. This study was designed to blind participants and outcome assessors. Patients were informed that they would be treated with either classical EA, non-classical EA, or usual care. To obtain the blinding of participants, a Streitberger’s method [24] was used. For both the EA group and the sham EA group, a plastic ring was banded to the skin surface. Then a seemingly identical needle was inserted between the ring [25]. The Streitberger placebo needle appeared to be the same as the acupuncture needle used in the EA group. With a blunt tip, the length of the needle shaft seen outside was gradually shortened without penetration, resulting in a visual effect of penetration to the skin [26]. The new blinding index [27] and treatment credibility scale [28] were assessed at the end of the first treatment and the last treatment. Participants were asked to answer which treatment was applied, including an answer of “don’t know”. Moreover, questions to measure treatment credibility, including “how confident do you feel that this treatment can alleviate cognitive impairment?”; “how logical does this treatment seem to you?”; “how confident would you be in recommending this treatment to a friend who suffer from cognitive impairment?”; and “how successful do you think this treatment would be in alleviating other complaints?”, were asked. Patients received each intervention in an environment similar to the real clinical setting. Independent and qualified clinical psychologists performed neuropsychological assessments throughout the study period. Since they were not in charge of the intervention, blinding was fully maintained for outcome assessors.

### 2.5. Interventions

#### 2.5.1. Electroacupuncture (EA)

A qualified Korean medicine doctor with clinical experience of more than two years performed EA for 24 sessions over 12 weeks (2 times a week). The skin in the treatment area was first sterilized with an isopropyl alcohol skin wipe. Sterile stainless steel acupuncture needles (0.25 mm in diameter and 25 mm in length, Asiamed, Suhl, Germany) were then inserted 5–25 mm into 14 classical acupoints, including GV20, GV24, EX-HN1, bilateral ST36, and bilateral KI3 (Figure 1a). Electric stimulation was applied at bilateral PC6 and HT7 [29] using an electroacupuncture device (ES-160, ITO Co. Ltd.) at a frequency of 3 Hz for 30 min. The intensity of the stimulation was adjusted to 80% of the intensity at which participants perceived electric stimulation. EA treatment was established using guidelines of the revised Standards for Reporting Interventions in Clinical Trial of Acupuncture (STRICTA) [30].

#### 2.5.2. Sham Electroacupuncture (Sham EA)

Sham EA was performed using the Streitberger placebo needle (Asiamed, Suhl, Germany) with a blunt tip that would not penetrate the skin [24]. The overall treatment process was applied the same as EA by a qualified Korean medicine doctor. Fourteen non-classical acupoints located in the upper limbs and lower limbs were selected to avoid the location-specific effect of EA. We chose sham acupoint locations not sharing the dermatomes with acupoints used in the EA group [31]. UL1 is on the anterolateral aspect of the arm at the prominence of the biceps brachii muscle belly. UL2 is located on the upper limb 2 cm superior to UL1. UL3 is on the anterior aspect of the elbow 5 cm inferior to the cubital crease and 1 cm lateral from the midline. UL4 is located on the upper limb 2 cm superior to UL3. LL1 is on the superior part of the 1/3 medial aspect of the tibia. LL2 is located on the lower limb 2 cm inferior to LL1. LL3 is located on the lower limb 2 cm inferior to LL2 (Figure 1b). Sham electric stimulation was applied with light and sound stimulation at a frequency of 3 Hz without a current flow.

#### 2.5.3. Usual Care for MCI Patients

All participants received education to enhance their cognitive function at every visit with a brochure. At the first visit, a qualified Korean medicine doctor conducted an education program for 10 to 15 min individually based on the brochure. At every visit, the progress of cognitive problems was checked for about 5 to 10 min. The education program was composed of basic information on MCI, lifestyle management to prevent cognitive disorders, and exercise guidelines with detailed pictures and descriptions to stimulate the facial/cervical region and to enhance whole body circulation. The education program and the brochure were developed based on previous clinical guidelines for dementia in Korea and the report by the international working group on mild cognitive impairment [32].

### 2.6. Outcomes

This pilot trial mainly focused on exploring potential effects of EA on various cognitive function domains. A detailed neuropsychological assessment battery of the Seoul neuropsychological screening battery II (SNSB-II) was examined at baseline, after a 12-week treatment (post-treatment), and at the 24-week time point of the follow-up assessment. It was composed of various validated cognitive tests to provide T scores for five cognitive function domains (attention, language and related functions, visuospatial functions, memory, and frontal/executive functions) [33,34] adjusted by age and education level. The T score of each cognitive function domain was determined based on various tests included in the battery. The battery consisted of the Digit Span Test (DST) forward and backward score [35] for the attention domain; Comprehension and Repetition score and Korean-Boston Naming Test (K-BNT) [36] for the language domain; Rey Complex Figure Test (RCFT) Copy score [37] and Clock drawing test (CDT) for the visuospatial domain; Rey Complex Figure Test (RCFT) [37] and Seoul Verbal Learning Test (SVLT) [38] for the memory domain; and Go-No Go, Controlled Oral Word Association Test (K-COWAT) [39], Korean-Color Word Stroop Test (K-CWST) [40], Digit Symbol Coding (DSC), and Korean-Trail Making Test-Elderly’s version (K-TMT) for the frontal/executive function domain. 

The Alzheimer’s disease assessment scale cognitive subscale-11 items (ADAS-Cog-11) [41,42] and MoCA-K [17,23] were examined at baseline, every four weeks during the treatment period at 4-, 8-, and 12-week time points, and at the 24-week time point of the follow-up assessment. The Hospital Anxiety and Depression Scale (HADS) [43,44] and short version of geriatric depression scale (SGDepS) [45] were measured at baseline, post-treatment, and at the follow-up assessment. Additionally, the Patient Global Impression of Change (PGIC) was examined at the post-treatment and follow-up assessment. The following question was asked to the participants: “How much did cognitive impairment improve compared to before participation in the study?”. Participants were asked to choose answers among “very much improved”, “much improved”, “minimally improved”, “no change”, “minimally worse”, “much worse”, and “very much worse”. Adverse events were carefully recorded during the study. Symptoms, severity, and causality of adverse events were documented. Laboratory tests were done at the baseline and post-treatment (12 weeks).

### 2.7. Statistical Analysis

This was a pilot clinical study to explore the effectiveness, safety, and feasibility of electro-acupuncture for mild cognitive impairment. For the purpose of this study, the sample size was determined to be 16 for each group considering that the standardized effect size of EA for MCI would be medium [46]. A total of 60 participants were planned to be recruited, accounting for an expected dropout rate of 20%. Adjustment for multiple comparisons of five cognitive domains was not considered in the sample size calculation, which was one limitation of this study.

Continuous variables are presented as the mean (standard deviation), and categorical variables are presented as a frequency (%). Statistical analyses for clinical outcomes were performed using the full analysis set (FAS) of the population, which included a population as similar to intent-to-treat (ITT) as possible. Participants who did not meet inclusion and exclusion criteria were excluded from the FAS. Per protocol (PP) analysis of clinical outcomes was carried out supplementarily. Safety analysis was performed for participants who received the intervention and safety evaluation more than once. All analyses were performed using SAS Version 9.4 with a significance level of 5% and two-sided tests. This pilot trial was designed as a three-arm study. There were two null hypotheses: (1) there was no difference in T scores on the SNSB cognitive function domain between the EA group and the usual care group at the post-treatment; and (2) there was no difference in T scores on the SNSB cognitive function domain between the EA group and the sham EA group at the post-treatment. A fixed sequence procedure was applied. If the previous null hypothesis was rejected, a subsequent hypothesis was then tested [47]. The least square mean difference between two groups was calculated by the analysis of covariance (ANCOVA) with the baseline score as the covariate and group as a fixed factor. Missing data were replaced by multiple imputations. No adjustment was made for multiple comparisons. Therefore, outcomes should be interpreted as exploratory.

## 3. Results

### 3.1. Participant Flow

Of 166 participants screened for eligibility, 106 were excluded due to not meeting the inclusion and exclusion criteria (*n* = 98) or declining to participate (*n* = 8). Most participants were excluded because of not meeting the criteria of MCI (*n* = 64) and high depression subscale in HADS (*n* = 35). Sixty participants were included and randomly allocated to each group in this trial (Figure 2). Twenty patients were assigned to each group (the EA group, the sham EA group, and the usual care group). During the treatment period, 11 participants declined to participate and discontinued the study, and two serious adverse events of admission not related to the intervention occurred. Two participants were found not to meet the inclusion criteria whose MoCA-K score was 23 after the randomization. Thus, they were excluded from the study. There were no notable differences in demographic characteristics at baseline among the three groups (Table 1). The subtype of MCI was judged based on the cognitive domain T score on the SNSB. Patients with a T score <40 for a domain were judged as having impairment in that domain. There were 31 (53.4%) patients with amnestic MCI and 19 (32.8%) patients with non-amnestic MCI. Eight (13.8%) patients had unspecified MCI with the above method. Their MoCA scores were below 23. They were included in this study, although they did not show impairment based on the T score on the SNSB-II.

From November to December 2016, 7 participants were recruited from one hospital. From December 2016 to August 2018, 53 participants were recruited from another hospital. Recruitment rates were 3.5 participants/month and 2.5 participants/month, respectively. Of 17 and 149 screened participants, 7 (41.2%) and 53 (35.6%) participants were enrolled from the two centers, respectively.

### 3.2. Effect of EA on Five Cognitive Function Domains

Results for the five cognitive function domains after the treatment are presented in Figure 3 and Table 2. Among the five cognitive domains of the SNSB, the visuospatial function domain score showed a difference between the EA group and the usual care group post-treatment. The visuospatial function T score in the EA group was increased from 36.40 ± 13.43 at baseline to 45.78 ± 13.20 (*p* = 0.0224) at the post-treatment assessment. In the usual care group, it changed from 35.72 ± 14.52 to 35.62 ± 15.15 (*p* = 0.8802). The LS mean difference of visuospatial function T score between the EA and the usual care group at post-treatment assessment was 10.16 (95% CI: 1.14, 19.18) (*p* = 0.0283). The effect size of EA compared to usual care for the visuospatial function T score calculated as the Cohen’s *d* was 0.72 (95% CI: 0.08, 1.35). The frontal/executive function T score was increased from 41.35 ± 10.93 to 49.29 ± 9.71 after 12 weeks of the treatment period in the EA group (*p* = 0.0100). In the usual care group, it changed from 42.65 ± 11.03 to 44.45 ± 9.32 (*p* = 0.4917). The LS mean difference of frontal/executive function T score between the EA and the usual care group at post-treatment assessment was 4.83 (95% CI: −1.30, 10.97) (*p* = 0.1190) with a Cohen’s *d* value of 0.51 (95% CI: −0.14, 1.15). For other cognitive domains, there was no noticeable difference between the EA group and the usual care group.

### 3.3. Effects of EA on ADAS-cog-11 and MoCA

Table 3 and Appendix A present results of the ADAS-cog-11 and MoCA after treatment. The ADAS-cog score was gradually decreased during the treatment period. It then slightly increased at the follow-up assessment in all three groups. There was no significant difference in the ADAS-cog between the EA group and the usual care group at the post-treatment assessment. The MoCA score was increased from the baseline after treatment in all three groups. There was no significant difference in its change among the three groups. For the subdomain of the MoCA, the visuospatial/executive score at week 8 was higher in the EA group than that in the usual care group (LS mean difference: 0.92 (95% CI: 0.17, 1.66), Cohen’s *d* = 0.81, *p* = 0.0170). It also tended to be higher than that in the sham EA group at week 8 (LS mean difference 0.66 (95% CI: −0.10, 1.41), Cohen’s *d* = 0.58, *p* = 0.0853).

### 3.4. Effects of EA on HADS and SGDepS

Table 4 presents results of the HADS and SGDepS after treatment. They showed no difference between the EA group and the usual care group at the post-treatment assessment. In the EA group, the anxiety score of the HADS gradually decreased from 6.80 ± 3.87 to 4.14 ± 3.59 at follow-up (*p* = 0.0016). In the sham EA group and the usual care group, it changed from 7.89 ± 3.11 to 6.55 ± 3.88 (*p* = 0.2797) and from 8.26 ± 5.01 to 6.57 ± 3.68 (*p* = 0.1954), respectively. The depression score of the HADS and SGDepS seemed to be decreased after treatment in the EA group. At the follow-up, the SGDepS score was lower in the EA group than in the usual care group (LS mean difference: −2.50 (95% CI: −4.79, −0.21), Cohen’s *d* = 0.67, *p* = 0.0334).

### 3.5. Effect of EA on PGIC

Figure 4 presents results of the PGIC after treatment. At the post-treatment assessment, 37.5%, 28.57%, and 29.41% of participants in the EA group, the sham EA group, and the usual care group reported that their symptoms were much/very much improved, respectively. At the follow-up assessment, 60%, 15.38%, and 25% of participants in the EA group, the sham EA group, and the usual care group reported that their symptoms were much/very much improved, respectively, showing no significant difference among the three groups (*p* = 0.687 at week 12 and *p* = 0.101 at week 24).

### 3.6. Adverse Events

All adverse events were carefully reported during the study. The most frequent adverse event was an upper respiratory infection, which was unlikely to be related to interventions. In the EA group, 7 (1.7%) cases of adverse events were possibly or probably related to the intervention, including headache, bruising, and pruritus in a total of 407 times of the EA intervention. In the sham EA group, 1 (0.3%) case of an adverse event was possibly or probably related to the intervention in a total of 371 times of the sham EA intervention. All adverse events that were probably and possibly related to EA and sham EA were mild and recovered naturally. Two severe adverse events occurred in the EA group and the sham EA group. They were definitely not related to the intervention. One was due to a traffic accident in the EA group, and the other was due to an accidental right knee bone fracture in the sham EA group (Table 5).

### 3.7. Blinding Test

A blinding test was done twice for the EA group and the sham EA group after the first treatment period (week 0) and at the end of the treatment period (week 12). At the first blinding test, the new blinding index was 0.7 (0.499, 0.901) for the EA group and −0.55 (−0.808, −0.292) for the sham EA group. At the second blinding test, the blinding index was 0.875 (0.713, 1.037) for the EA group and −0.667 (−1.021, −0.313) for the sham EA group. Participants in both groups thought that they received classical electro-acupuncture, meaning that the blinding was successful. The blinding was successful throughout the study. It was maintained from the first treatment to the last treatment (Table 6). The credibility of treatment was also measured at the same time, showing no significant difference between the EA group and the sham EA group, meaning that the blinding was successful (Appendix A).

## 4. Discussion

The aim of this pilot study was to explore which cognitive domain could be improved by the EA. Due to the small sample size, it was not enough to conclude the effect of EA on various cognitive functions. According to the results of our pilot study, the estimated effect size (Cohen’s *d*) of EA on visuospatial function compared to usual care was 0.72, indicating a medium effect [48]. Moreover, the effect size of EA on the frontal/executive domain compared to usual care was 0.51, also indicating a medium effect. A noticeable difference between the EA group and the sham EA group was not observed in this study.

Visuospatial dysfunction that relies on the parietal lobe is one of the prominent early features of Alzheimer’s disease [49]. Executive dysfunction may also contribute to the impairment of instrumental activities of daily living, which is important for patients’ and caregivers’ quality of life [50]. According to a large longitudinal aMCI cohort in Korea, frontal/executive dysfunction can increase the conversion rate from aMCI to dementia [15]. A previous study on healthy older adults showed that EA treatment on GV20, EX-HN1, GV24, LI4, HT7, PC6, ST36, LR3, SP6, KI3 can enhance the visuospatial/executive function score of MoCA compared to the control [51]. Another study on patients with vascular cognitive impairment reported that EA can improve the visuospatial/executive function score of MoCA compared to the control [52]. A randomized controlled trial in China showed that MCI patients treated with EA have a better improvement in the CDT score, a representative test for visuospatial function, than MCI patients treated with nimodipine [53]. Thus, improvements in visuospatial/executive function after the EA treatment could be expected [22].

There have been a few pilot studies on acupuncture for attention, language, and memory domains. In a previous pilot study of acupuncture for MCI patients, acupuncture at EX-HN1, EX-HN3, PC6, KI3, ST40, and LR3 improved the digit span test score, a representative test for attention [22]. In another study, stroke patients with impaired cognition who applied EA at PC6 and HT7 showed a tendency of higher memory function compared to the control group, which was not statistically significant [29]. Moreover, a systematic review summarized that acupuncture is effective in improving language function in patients with post-stroke aphasia [54]. In our pilot study, the estimated effect sizes of EA for the attention, language, and memory domains were relatively small. However, this pilot study was conducted with a limited sample size, and the 95% CI ranges of the estimated effect size were relatively wide. Therefore, in the future, with confirmative research with a sufficient sample size, there is a possibility that results will differ from those of this pilot study. Based on the results of this study, it will be possible to plan a confirmative trial with an appropriate sample size to find which cognitive function domains are specifically improved by EA.

There have been limited studies about the effects of EA and sham EA on cognitive impairment. In most previous randomized controlled trials of EA for cognitive disorder, active controls (e.g., nimodipine and donepezil) were selected for comparison while a sham control was rarely used [5,55]. In our pilot study, we used the Streitberger placebo needle [24] that would not penetrate the skin at non-classical acupoints without applying an electric stimulation. Blinding was assessed as successful throughout the study, considering that most participants in both groups thought that they received classical EA. The credibility of treatment between the two groups was not statistically different. EA has three active components [56]: acupuncture needle location, depth of acupuncture needle insertion, and electric stimulation. First, the acupoint location used in this study—PC6 and HT7—might not have a specific effect on EA stimulation in MCI patients. Another previous study reported that KI3 has a different effect on intrinsic brain activity compared to the sham acupoint in patients with mild cognitive impairment [21]. Moreover, GV20, which has been commonly used in other trials of MCI, may have a potential specific effect. Limited studies have compared GV20 with sham acupoints for MCI patients. One trial using GV20 for depressed patients reported a location-specific effect of GV20 [57]. Second, the Streitberger placebo needle that could touch the skin without penetrating the skin was used in this study. For pain management, there have been reports that EA with skin penetration is superior to sham EA without skin penetration [58]. Meanwhile, a previous study on MCI patients reported that deep needle insertion on KI3 shows a different effect on the brain compared to superficial insertion [59], which required further research about the specific effect of the depth of acupuncture insertion for MCI. Third, 3 Hz of electric stimulation was used in this study. Different methods of EA stimulation may be required. An in vivo study showed that high-frequency electric stimulation (50 Hz) was superior to ameliorate cognitive impairments in a rat model compared to low (2 Hz) or medium (30 Hz) frequency electric stimulation [60].

In our pilot trial, the SGDepS score, which measures geriatric depression symptoms, seemed to show improvement after the EA treatment. We excluded patients who were having cognitive impairment due to depression. The baseline mean HADS-D score was 5.90, which was under the 8-point of cut-off value for screening depression [44]. However, the co-morbid prevalence of MCI and depression has been reported to be high, ranging from 25% to 50%. Managing geriatric depressive symptoms in MCI patients is also important [61]. EA has been reported to have a potential for relieving depression [62,63,64]. Our pilot study results suggest that EA may improve depressive symptoms in MCI patients. Further studies are needed to explore effects of EA on both depressive symptoms and cognitive decline among elderly people.

Mild adverse events that were reported mainly included bruising after EA treatment. In a previous study, hematoma and bleeding occurred in 3.19% and 1.38% of patients who received acupuncture treatment, respectively [65]. In another prospective observational study, 6.14% of acupuncture-treated patients experienced bleeding/hematoma, and 81.4% of those adverse events did not need treatment [66]. In our study, bruising occurred four times (0.98%) in a total of 407 times of EA treatment. Electric stimulation was applied at PC6 and HT7 located in the ventral forearm [67] and acupuncture points on the cranial skin and superficial fascia known to have abundant blood circulation [68]. This might be a reason for the high frequency of bruising in the EA group. Based on our inclusion criteria, elderly participants with an age over 40 years but under 80 years were included. The mean age of participants was over 60 years. It has been reported that the older the patient, the higher the risk of side effects [68]. When EA treatment is applied to elderly MCI patients, the common risk of minor bleeding and bruising may be considered. In every case of our pilot trial, those adverse events were minor. They did not need additional treatment other than close observation.

Various acupuncture points were used for cognitive disorders. In our trial, bilateral PC6 and HT7 were used for electric stimulation based on a previous study about the effect of EA on the cognitive function of stroke patients [29]. In other clinical trials of EA for MCI that excluded vascular MCI (vMCI), EA stimulation was mainly applied on the head and neck region, including EX-HN1, GV20, GV24, and GB20 [4]. Frequently used acupoints for MCI, including GV24, GV20, EX-HN1, and KI3, were also used in our study. However, electric stimulation was not applied at these acupoints. In animal studies, EA stimulation was applied on the head and neck region or KI3 on the ankle region [69,70,71,72]. Further studies are required to test the potential effect of electric stimulation on acupoints located in the head and neck region for improving cognitive decline.

This study has several limitations. First, considering the Bonferroni correction for multiple comparisons of the five cognitive function domains, the sample size should be larger. After the Bonferonni correction, a *p*-value less than 0.01 indicated statistical significance. Results of our study were insufficient to conclude the effect of EA on cognitive functions. A large-scale clinical study is needed. Second, EA treatment twice a week might not have been sufficient enough to be effective in improving cognitive functions. In a previous study, EA treatment was performed three times a week [19]. Third, the drop-out ratio was relatively high because the treatment period and follow-up period were long. PP analysis was also conducted. Results are presented in Appendix A, showing a similar tendency to results of the FAS analysis. Another analysis adjusting the age, sex, education year, and site also showed similar results (Appendix A).

This pilot study also aimed to evaluate the feasibility of a massive clinical trial. Sixty participants were recruited from two Korean medical university hospitals during about two years. Approximately 2–3 participants were recruited per month. The screening failure rate was 66.85% mainly because of not meeting the inclusion criteria (MoCA-K score over 22 or HADS score over 11). The drop-out rate was 25.00% at the follow-up period. The assessment for each participant took more than two hours. Modifying the assessment to be simpler and briefer would lower the drop-out rate. If the visuospatial function T score of the SNSB-II is used as the primary outcome in further research, the required sample size is estimated to be 31 per group based on our pilot study result. The ADAS-cog-11 did not appear to be appropriate to measure the effect of EA for patients with MCI. The ADAS-cog-13 with additional tests for MCI [41,73,74] should be considered. Regarding sham control, it would be better to find ways to increase the specific effect of EA by improving the electric stimulation method and acupoint selection for cognitive impairment.

## 5. Conclusions

In this pilot trial, we could see a possible but not statistically significant effect of EA on visuospatial function in patients with MCI. From our results, the effect size of EA on visuospatial function was estimated to be medium. It was not enough to conclude the effect of EA on various cognitive functions based on our study. Large-scale clinical trials are needed to confirm effects of EA on cognitive functions. Results of this pilot trial may provide useful information for further clinical trials of EA for MCI patients.

## Figures and Tables

**Figure 1 brainsci-11-00756-f001:**
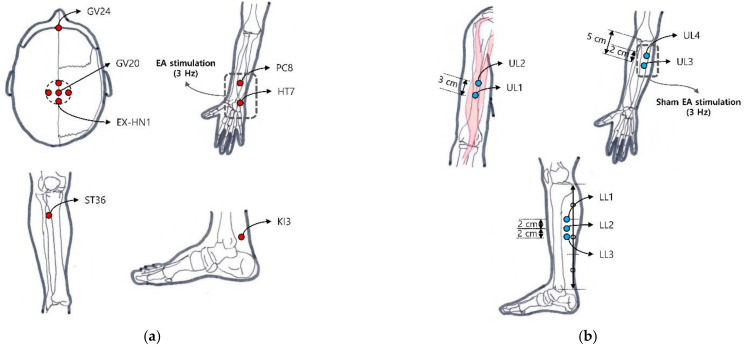
Locations of classical acupoint and non-classical acupoint: (**a**) GV24, GV20, EX-HN1, bilateral ST36, PC8, HT7, KI3 were selected as classical acupoints in the EA group; (**b**) On the other hand, four bilateral points located in upper limbs and three bilateral points located in lower limbs were selected for the sham electro-acupuncture group. Sham EA stimulation was applied with light and sound stimulation at a frequency of 3Hz without electric stimulation.

**Figure 2 brainsci-11-00756-f002:**
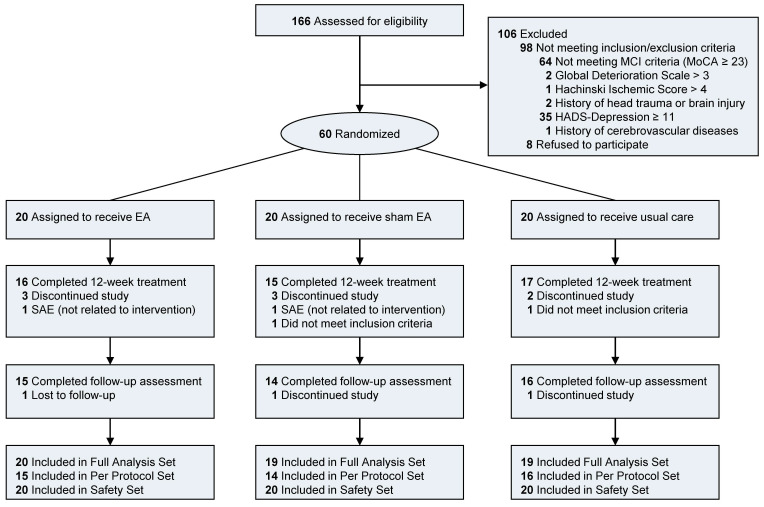
Flow diagram of participants in the pilot trial of EA for mild cognitive impairment. Number of participants not meeting each inclusion/exclusion criteria were counted individually. There were six participants who did not meet the MCI criteria with higher HADS-D than 11. One participant did meet the MCI criteria with GDS higher than 3. Two serious adverse events (SAE) occurred. Both were definitely not related to interventions. Two participants did not meet the inclusion criteria after the randomization. Thus, they were excluded from the Full Analysis Set (FAS).

**Figure 3 brainsci-11-00756-f003:**
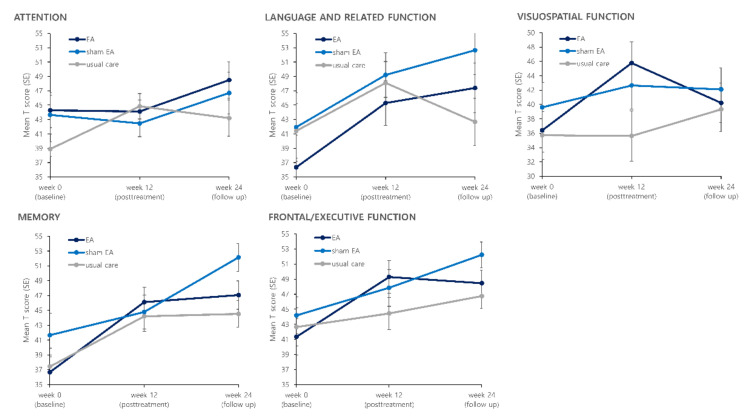
Effects of treatment on five cognitive domains of Seoul Neuropsychological Screening Battery II (SNSB-II).

**Figure 4 brainsci-11-00756-f004:**
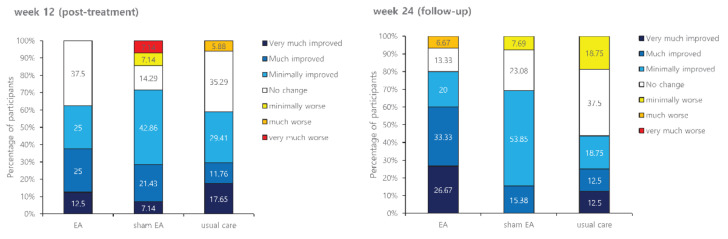
Effects of treatment on Patient Global Impression of Change (PGIC).

**Table 1 brainsci-11-00756-t001:** Baseline Characteristics of Participants.

	EA Group (*n* = 20)	Usual Care Group (*n* = 19)	Sham EA Group (*n* = 19)	*p*-Value
Sex				
Male	2 (10.0%)	2 (10.5%)	2 (10.5%)	0.998
Female	18 (90.0%)	17 (89.5%)	17 (89.5%)	
Age, y	64.1 (6.6)	62.9 (6.5)	64.1 (8.4)	0.849
Education, y	9.0 [6.0]	9.0 [6.0]	9.0 [6.0]	0.939
≤6 years	5 (25.0%)	6 (31.6%)	6 (31.6%)	0.813
7~9 years	7 (35.0%)	5 (26.3%)	5 (26.3%)	
10~12 years	6 (30.0%)	8 (42.1%)	6 (31.6%)	
≥13 years	2 (10.0%)	0 ( 0.0%)	2 (10.5%)	
Age at onset of cognitive problems, y	62.7 (6.5)	59.1 (8.1)	62.4 (9.9)	0.333
Duration of cognitive problems, y	2.6 [2.2]	3.5 [5.5]	2.4 [1.8]	0.244
Medical History				
Hypertension	4 (20.0%)	5 (26.3%)	10 (52.6%)	0.073
Diabetes mellitus	2 (10.0%)	2 (10.5%)	3 (15.8%)	0.831
Hyperlipidemia	6 (30.0%)	2 (10.5%)	6 (31.6%)	0.238
Employment				
Employed	6 (30.0%)	12 (63.2%)	8 (42.1%)	0.110
Unemployed	14 (70.0%)	7 (36.8%)	11 (57.9%)	
HADS				
Depression	5.2 (2.7)	7.0 (2.9)	6.4 (2.6)	0.138
Anxiety	6.8 (3.9)	8.3 (5.0)	7.9 (3.1)	0.508
MoCA	19.6 (2.1)	19.4 (2.6)	19.2 (3.1)	0.840
MMSE	25.9 (2.5)	26.0 (1.4)	26.2 (1.8)	0.850
CDR-SB	0.5 [0.3]	0.5 [0.5]	0.5 [0.3]	0.604
GDS				
GDS 2	10 (50.0%)	8 (42.1%)	11 (57.9%)	0.623
GDS 3	10 (50.0%)	11 (57.9%)	8 (42.1%)	
ADAS-cog-11				
Hachinski ischemic score	2.0 [1.0]	1.0 [1.0]	2.0 [1.0]	0.253
MCI subtypes				
aMCI (single)	1 (5.0%)	0 (0.0%)	0 (0.0%)	0.159
aMCI (multiple)	9 (45.0%)	13 (68.4%)	8 (42.1%)	
naMCI (single)	4 (20.0%)	3 (15.8%)	3 (15.8%)	
naMCI (multiple)	5 (25.0%)	2 (10.5%)	2 (10.5%)	
Unspecified	1 (5.0%)	1 (5.3%)	6 (31.6%)	
Acupuncture Expectancy Score	4.4 [1.6]	4.0 [0.6]	3.5 [2.0]	0.314

Data are presented as n (%), mean (sd), or median [iqr]. aMCI, amnestic mild cognitive impairment; EA, electroacupuncture; HADS, hospital anxiety and depression scale; MCI, mild cognitive impairment; MoCA, Montreal Cognitive Assessment; MMSE, Mini-Mental State Examination; CDR-SB, Clinical Dementia Rating-Sum of Boxes; GDS, global deterioration scale; ADAS-cog-11, Alzheimer’s disease assessment scale-cognitive subscale-11 items; naMCI, non-amnestic mild cognitive impairment.

**Table 2 brainsci-11-00756-t002:** Effects of 12-week EA Treatment on T scores of Five Cognitive Function Domains Calculated with Cognitive Tests Included in Seoul Neuropsychological Screening Battery II (SNSB-II).

Cognitive Domains	EA Group	Usual Care Group	Mean Difference ^1^	Cohen’s d	*p-* Value ^1^	Sham EA Group	Mean Difference ^1^	Cohen’s d	*p*- Value ^1^
**Attention**									
Baseline	44.29 (10.73)	38.89 (4.51)				43.66 (11.73)			
Post-treatment	44.12 (7.03)	44.84 (7.65)	−0.73 (−5.56, 4.11)	−0.10 (−0.76, 0.56)	0.7628	42.46 (7.97)	1.66 (−3.27, 6.58)	0.22 (−0.44, 0.88)	0.4993
difference ^2^	0.84 (−3.55, 5.24)	4.19 (1.02, 7.36)				−0.51 (−5.47, 4.46)			
*p*-value ^2^	0.6901	0.0139				0.8287			
Follow-up	48.50 (11.43)	43.20 (10.87)	5.30 (−2.10, 12.69)	0.47 (−0.19, 1.14)	0.1554	46.70 (12.58)	1.80 (−6.16, 9.76)	0.15 (−0.51, 0.81)	0.6478
difference ^2^	4.93 (−1.30, 11.16)	3.06 (−2.50, 8.62)				3.53 (−3.67, 10.74)			
*p*-value ^2^	0.1113	0.2587				0.3041			
**Language function**									
Baseline	36.34 (20.16)	41.41 (16.57)				41.94 (20.78)			
Post-treatment	45.29 (13.76)	48.10 (12.98)	−2.80 (−11.13, 5.52)	−0.21 (−0.83, 0.41)	0.5007	49.21 (13.62)	−3.92 (−13.12, 5.27)	−0.29 (−0.96, 0.39)	0.3920
difference ^2^	8.12 (−1.18, 17.42)	7.06 (−0.89, 15.01)				7.78 (−3.48, 19.03)			
*p*-value ^2^	0.0825	0.0778				0.1616			
Follow-up	47.40 (15.47)	42.68 (14.16)	4.71 (−5.06, 14.49)	0.32 (−0.34, 0.98)	0.3346	52.66 (14.73)	−5.26 (−15.40, 4.88)	−0.35 (−1.02, 0.32)	0.2991
difference ^2^	9.23 (0.28, 18.17)	2.10 (−5.92, 10.12)				11.82 (3.75, 19.88)			
*p*-value ^2^	0.0441	0.5844				0.0072			
**Visuospatial function**									
Baseline	36.40 (13.43)	35.72 (14.52)				39.60 (14.99)			
Post-treatment	45.78 (13.20)	35.62 (15.15)	10.16 (1.14, 19.18)	0.72 (0.08, 1.35)	0.0283	42.65 (14.45)	3.13 (−5.70, 11.96)	0.23 (−0.41, 0.87)	0.4791
difference ^2^	9.08 (1.46, 16.70)	−0.65 (−9.84, 8.54)				3.93 (−3.89, 11.75)			
*p*-value ^2^	0.0224	0.8802				0.2972			
Follow-up	40.22 (12.27)	39.33 (13.45)	0.89 (−7.53, 9.32)	0.07 (−0.59, 0.72)	0.8305	42.10 (12.97)	−1.87 (−9.95, 6.20)	−0.15 (−0.79, 0.49)	0.6418
difference ^2^	3.43 (−3.25, 10.12)	2.90 (−5.50, 11.29)				3.62 (−3.18, 10.42)			
*p*-value ^2^	0.2908	0.4686				0.2688			
**Memory**									
Baseline	36.67 (10.38)	37.43 (10.87)				41.66 (12.77)			
Post-treatment	46.11 (8.90)	44.20 (8.75)	1.91 (−3.74, 7.56)	0.22 (−0.42, 0.86)	0.4988	44.78 (9.93)	1.33 (−4.84, 7.50)	0.14 (−0.51, 0.80)	0.6655
difference ^2^	8.25 (3.77, 12.73)	6.06 (1.68, 10.44)				5.07 (−0.69, 10.84)			
*p*-value ^2^	0.0013	0.0101				0.0796			
Follow-up	47.06 (8.59)	44.52 (7.76)	2.54 (−2.68, 7.75)	0.31 (−0.33, 0.95)	0.3315	52.15 (8.25)	−5.09 (−10.23, 0.05)	−0.60 (−1.21, 0.01)	0.0524
difference ^2^	9.34 (5.45, 13.22)	6.47 (2.02, 10.91)				12.22 (6.60, 17.84)			
*p*-value ^2^	0.0002	0.0073				0.0004			
**Frontal/executive function**									
Baseline	41.35 (10.93)	42.65 (11.03)				44.18 (10.61)			
Post-treatment	49.29 (9.71)	44.45 (9.32)	4.83 (−1.30, 10.97)	0.51 (−0.14, 1.15)	0.1190	47.85 (10.69)	1.44 (−5.53, 8.41)	0.14 (−0.54, 0.82)	0.6766
difference ^2^	7.18 (1.99, 12.37)	1.77 (−3.59, 7.13)				4.49 (−1.09, 10.06)			
*p*-value ^2^	0.0100	0.4917				0.1039			
Follow-up	48.46 (7.62)	46.74 (7.03)	1.72 (−3.10, 6.54)	0.23 (−0.42, 0.89)	0.4725	52.23 (7.41)	−3.76 (−8.90, 1.38)	−0.50 (−1.18, 0.18)	0.1451
difference ^2^	6.44 (1.40, 11.47)	4.07 (0.11, 8.02)				8.78 (4.59, 12.97)			
*p*-value ^2^	0.0158	0.0446				0.0006			

Data are presented as mean (sd) or mean (95% CI); EA, electroacupuncture; ^1^ Least squares mean difference and *p*-value by Analysis of covariance (ANCOVA); ^2^ Mean difference and *p*-value by paired t-test.

**Table 3 brainsci-11-00756-t003:** Effects of 12-week EA Treatment on MoCA and ADAS-cog-11.

Cognitive Domains	EA Group	Usual Care Group	Mean Difference ^1^	Cohen’s d	*p*- Value ^1^	Sham EA Group	Mean Difference ^1^	Cohen’s d	*p*- Value ^1^
**MoCA**									
Baseline	19.65 (2.06)	19.42 (2.61)				19.16 (3.06)			
Week 4	23.28 (3.20)	23.37 (3.28)	−0.09 (−2.18, 1.99)	−0.03 (−0.67, 0.62)	0.9309	23.36 (3.19)	−0.09 (−2.14, 1.97)	−0.03 (−0.67, 0.62)	0.9341
difference ^2^	3.72 (2.57, 4.87)	3.95 (1.74, 6.16)				4.11 (2.41, 5.80)			
*p*-value ^2^	<0.0001	0.0017				0.0001			
Week 8	24.56 (3.10)	24.14 (3.11)	0.42 (−1.59, 2.42)	0.13 (−0.51, 0.78)	0.6787	23.55 (3.23)	1.00 (−1.02, 3.03)	0.32 (−0.32, 0.96)	0.3242
difference ^2^	4.98 (3.67, 6.30)	4.72 (2.76, 6.69)				4.31 (2.43, 6.20)			
*p*-value ^2^	<0.0001	0.0001				0.0002			
Post-treatment	24.44 (2.68)	24.89 (2.65)	−0.45 (−2.15, 1.25)	−0.17 (−0.81, 0.47)	0.5960	25.25 (2.94)	−0.80 (−2.69, 1.09)	−0.29 (−0.96, 0.39)	0.3933
difference ^2^	4.86 (3.77, 5.96)	5.47 (3.66, 7.28)				6.01 (4.19, 7.83)			
*p*-value ^2^	<0.0001	<0.0001				<0.0001			
Follow-up	25.24 (3.34)	24.34 (3.24)	0.90 (−1.17, 2.97)	0.27 (−0.36, 0.90)	0.3859	25.42 (3.41)	−0.18 (−2.31, 1.95)	−0.05 (−0.68, 0.58)	0.8628
difference ^2^	5.64 (4.11, 7.18)	4.92 (2.98, 6.86)				6.21 (4.02, 8.39)			
*p*-value ^2^	<0.0001	0.0001				<0.0001			
**ADAS−cog−11**									
Baseline	12.30 (5.32)	12.74 (5.63)				11.32 (3.07)			
Week 4	10.34 (3.06)	10.05 (3.08)	0.29 (−1.70, 2.28)	0.09 (−0.55, 0.74)	0.7694	10.11 (3.01)	0.23 (−1.73, 2.19)	0.08 (−0.57, 0.72)	0.8130
difference ^2^	−1.89 (−3.94, 0.17)	−2.44 (−4.68, −0.20)				−1.53 (−3.27, 0.22)			
*p*-value ^2^	0.0693	0.0346				0.0819			
Week 8	9.21 (2.78)	9.59 (2.75)	−0.38 (−2.16, 1.40)	−0.14 (−0.78, 0.50)	0.6684	8.91 (2.89)	0.30 (−1.51, 2.10)	0.10 (−0.53, 0.74)	0.7431
difference ^2^	−3.03 (−4.93, −1.13)	−2.93 (−5.11, −0.75)				−2.69 (−4.67, −0.71)			
*p*-value ^2^	0.0038	0.0116				0.0112			
Post-treatment	8.94 (3.15)	8.17 (3.05)	0.77 (−1.24, 2.78)	0.25 (−0.40, 0.90)	0.4433	8.51 (3.31)	0.43 (−1.67, 2.52)	0.13 (−0.52, 0.78)	0.6813
difference ^2^	−3.31 (−5.77, −0.85)	−4.40 (−6.74, −2.05)				−3.03 (−5.01, −1.04)			
*p*-value ^2^	0.0114	0.0011				0.0056			
Follow-up	9.69 (3.93)	9.11 (3.86)	0.58 (−1.87, 3.03)	0.15 (−0.48, 0.78)	0.6364	9.08 (4.15)	0.61 (−2.04, 3.25)	0.15 (−0.50, 0.81)	0.6442
difference ^2^	−2.57 (−5.43, 0.29)	−3.49 (−6.30, −0.68)				−2.41 (−4.64, −0.18)			
*p*-value ^2^	0.0746	0.0182				0.0363			

Data are presented as mean (sd) or mean (95% CI). EA, electroacupuncture; MoCA, Montreal Cognitive Assessment; ADAS-cog, Alzheimer’s disease assessment scale cognitive subscale. ^1^ Least squares mean difference and *p*-value by Analysis of covariance (ANCOVA). ^2^ Mean difference and *p*-value by paired t-test.

**Table 4 brainsci-11-00756-t004:** Effects of 12-week EA Treatment on HADS and SGDepS.

	EA Group	Usual Care Group	Mean Difference ^1^	Cohen’s *D*	*p*- Value ^1^	Sham EA Group	Mean Difference ^1^	Cohen’s *D*	*p*- Value ^1^
**Anxiety score (HADS)**									
Baseline	6.80 (3.87)	8.26 (5.01)				7.89 (3.11)			
Post-treatment	5.68 (3.62)	6.07 (3.52)	−0.39 (−2.71, 1.92)	−0.11 (−0.76, 0.54)	0.7333	6.15 (3.91)	−0.48 (−2.93, 1.97)	−0.13 (−0.78, 0.52)	0.6955
difference ^2^	−1.64 (−3.36, 0.09)	−1.81 (−3.87, 0.25)				−1.58 (−3.62, 0.45)			
*p*-value ^2^	0.0612	0.0801				0.1148			
Follow-up	4.14 (3.59)	6.57 (3.68)	−2.43 (−4.83, −0.02)	−0.67 (−1.33, −0.01)	0.0482	6.55 (3.88)	−2.40 (−4.83, 0.02)	−0.64 (−1.30, 0.01)	0.0523
difference ^2^	−3.03 (−4.72, −1.35)	−1.41 (−3.63, 0.81)				−1.23 (−3.59, 1.12)			
*p*-value ^2^	0.0016	0.1954				0.2797			
**Depression score (HADS)**									
Baseline	5.25 (2.73)	7.00 (2.92)				6.42 (2.59)			
Post-treatment	4.96 (3.54)	5.62 (3.29)	−0.66 (−2.95, 1.63)	−0.19 (−0.86, 0.48)	0.5637	6.99 (3.50)	−2.03 (−4.27, 0.21)	−0.58 (−1.21, 0.06)	0.0747
difference ^2^	−0.86 (−2.50, 0.78)	−0.90 (−2.66, 0.86)				0.70 (−1.07, 2.47)			
*p*-value ^2^	0.2788	0.293				0.407			
Follow-up	4.50 (4.26)	6.04 (4.10)	−1.53 (−4.25, 1.18)	−0.37 (−1.02, 0.28)	0.2594	6.90 (3.89)	−2.40 (−4.98, 0.18)	−0.59 (−1.22, 0.04)	0.0673
difference ^2^	−1.14 (−3.31, 1.03)	−0.64 (−2.87, 1.59)				0.57 (−1.32, 2.46)			
*p*-value ^2^	0.2786	0.5484				0.5276			
**SGDepS**									
Baseline	4.30 (3.93)	4.21 (3.10)				4.00 (3.93)			
Post-treatment	2.81 (4.03)	3.57 (3.52)	−0.76 (−3.15, 1.63)	−0.20 (−0.83, 0.43)	0.5243	4.56 (3.65)	−1.75 (−4.27, 0.77)	−0.45 (−1.11, 0.20)	0.1671
difference ^2^	−1.42 (−3.61, 0.78)	−0.62 (−2.31, 1.07)				0.45 (−1.46, 2.36)			
*p*-value ^2^	0.1868	0.4447				0.6203			
Follow-up	2.17 (3.71)	4.67 (3.77)	−2.50 (−4.79, −0.21)	−0.67 (−1.28, −0.06)	0.0334	3.28 (3.46)	−1.11 (−3.38, 1.16)	−0.31 (−0.94, 0.32)	0.3302
difference ^2^	−2.07 (-4.27, 0.12)	0.48 (−1.38, 2.34)				−0.79 (−2.89, 1.30)			
*p*-value ^2^	0.0624	0.5816				0.4314			

Data are presented as mean (sd) or mean (95% CI). EA, electroacupuncture; HADS, Hospital Anxiety and Depression Scale; SGDepS, short version of geriatric depression scale. ^1^ Least squares mean difference and *p*-value by Analysis of covariance (ANCOVA). ^2^ Mean difference and *p*-value by paired t-test.

**Table 5 brainsci-11-00756-t005:** Adverse events (all causalities) reported during the study.

	EA Group (*n* = 20)	Usual Care Group (*n* = 20)	Sham EA Group (*n* = 20)
**Adverse events (Possibly, Probably related)**			
Headache	1	1	1
Bruise	4	0	0
Pruritus	2	0	0
**Severity of AEs**			
Mild	16	8	1
Moderate	1	0	0
Severe	1	1	0
**Causality of AEs**			
Definitely related	0	0	0
Probably related	6	0	0
Possibly related	1	1	1
Unlikely related	8	1	0
Definitely not related	3	7	0
Total number of participants with AE	10	5	1
Total number of AEs	20	9	1
Total number of AEs (intervention related)	7	1	1
Total number of SAEs	1	1	0
Total number of SAEs (intervention related)	0	0	0
Total number of interventions (or visits)	407	371	88

**Table 6 brainsci-11-00756-t006:** Blinding test.

Week 0	EA Group	Sham EA Group
Classical electro-acupuncture	14	12
Non-classical electro-acupuncture	0	1
Don’t know	6	6
New Blind Index (95% CI)	0.700 (0.499, 0.901)	−0.579 (−0.845, −0.313)
**Week 12**	**EA Group**	**Sham EA Group**
Classical electro-acupuncture	14	12
Non-classical electro-acupuncture	0	2
Don’t know	2	1
New Blind Index (95% CI)	0.875 (0.713, 1.037)	−0.643 (−1.019, −0.267)

## Data Availability

The data presented in this study are available on request from the corresponding author.

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
