# Peer review of "Feasibility and Effect of Electroacupuncture on Cognitive Function Domains in Patients with Mild Cognitive Impairment: A Pilot Exploratory Randomized Controlled Trial"

_brainsci, 2021, doi:10.3390/brainsci11060756_

Round 1

Reviewer 1 Report

I have to say that this article is very well presented, the methods are explained in excellent manner, the topic is quite interesting. However I find two major problems with the study: one sample size. The sample size is discrete and may be hindering the results. Second the authors go from trying to see MCI results to focusing in visuospatial exclusively. This is problematic as the title is. The title makes a question that the study can respond but it is not fair for the technique since it may appear that EA could only be valid for that specific domain when the truth is that sample size and other variables may be responsible for such results. For the benefit of the technique and to be more objective the title should be changed.

The discussion is too long and goes around to avoid the truth of the results. To bring animal models in this section is no needed at all. Authors should focus in the reality of the results. They found no significant differences and low size effects for all aspects studied except for maybe one. Then further research with larger samples could tell a different story.

Author Response

Thank you for giving us lots of valuable comments and the opportunity to improve our manuscript. Please see the attachment for our point-by-point responses.

Comment 1:

I have to say that this article is very well presented, the methods are explained in excellent manner, the topic is quite interesting. However, I find two major problems with the study: one sample size. The sample size is discrete and may be hindering the results.

Response 1:

Thank you for your valuable comments. The relatively small sample size is one of the limitations of our study. It was not sufficient to conclude the effect of electroacupuncture on various cognitive domains. We tried not to overestimate our pilot results in writing the manuscript. As a pilot study, this study hopefully provides basic data for a further large-scale clinical trial.

Comment 2:

Second the authors go from trying to see MCI results to focusing in visuospatial exclusively. This is problematic as the title is. The title makes a question that the study can respond but it is not fair for the technique since it may appear that EA could only be valid for that specific domain when the truth is that sample size and other variables may be responsible for such results. For the benefit of the technique and to be more objective the title should be changed.

Response 2:

Thank you for your valuable and important comment. Our study was a pilot trial, and we could not answer the question of original title. We revised the title to focus on the results of our pilot trial.

(Page 1, Title) Feasibility and Effect of Electroacupuncture on Cognitive Function Domains in Patients with Mild Cognitive Impairment: A Pilot Explanatory Randomized Controlled Trial

As a pilot study with relatively small sample size, we tried not to overestimate our results. Thank you for your valuable comments, and we revised the first paragraph of discussion as below.

(Page 14, line 391-392) The aim of this pilot study was to explore which cognitive domain could be improved by the EA. Due to small sample size, it was not enough to conclude the effect of EA on various cognitive functions. According to the results of our pilot study, the estimated effect size (Cohen’s d) of EA on visuospatial function compared to usual care was 0.72, indicating a medium effect.

Comment 3:

The discussion is too long and goes around to avoid the truth of the results. To bring animal models in this section is no needed at all. Authors should focus in the reality of the results. They found no significant differences and low size effects for all aspects studied except for maybe one. Then further research with larger samples could tell a different story.

Response 3:

Thank you for your valuable suggestion. We revised the discussion to focus on the reality of our results.

(Page 15, line 411-441) The mechanisms underlying the effect of EA on cognitive function have been reported in various animal models of AD. For example, bilateral EA stimulation at KI3 acupoint can attenuate microglia-mediated Aβ deposition in 5XRAD transgenic mice model of AD [1]. EA at GV20 and BL23 can reduce Aβ and p-Tau accumulation by activating PPAR-γ in an Aβ-injected AD rat model [2]. EA at governor vessel acupoints (GV20, GV29) can improve cognitive impairment by enhancing glucose metabolism in the hippocampus of APP/PS1 transgenic mice [3]. EA on GV20, GV26, and EX-HN3 can also increase the uptake rate of glucose in the hippocampi of AD mice [4]. Also, a systematic review of animal studies has summarized that acupuncture-related treatment might improve cognitive impairment by protecting synaptic damage, promoting cholinergic neural transmission, enhancing neurotrophin signaling, suppressing oxidative stress, and regulating glucose metabolism [5]. In a human fMRI study, EA can increase the functional connectivity within the default mode network which declines related to aging [6]. In another fMRI study, deep acupuncture insertion at KI3 can enhance functional connectivity in the hippocampus, amygdala, and insula of MCI patients compared to superficial acupuncture, suggesting that deep insertion of acupuncture might be important for treating cognitive disorders [7].

There have been a few pilot studies of acupuncture for attention, language, and memory domains. In a previous pilot study of acupuncture for MCI patients, acupuncture at EX-HN1, EX-HN3, PC6, KI3, ST40, and LR3 improved the digit span test score, a representative test for attention (Tan, 2017). In another study, stroke patients with impaired cognition who applied EA at PC6 and HT7 showed tendency of higher memory function compared to control group, which was not statistically significant (Chou, 2009). Also, a systematic review summarized that acupuncture is effective in improving language function in patients with post-stroke aphasia (Binlong Zhang, 2019). In our pilot study, the estimated effect size of EA for attention, language, and memory domain was relatively small. However, this pilot study was conducted with limited sample size, and the 95% CI ranges of estimated effect size were relatively wide. Therefore, in the future confirmative research with enough sample size, there is a possibility that results will differ from those of this pilot study. Based on the results of this study, it will be possible to plan a confirmative trial with an appropriate sample size to find which cognitive function domains are specifically improved by the EA.

Reviewer 2 Report

The manuscript titled “Which Cognitive Domain is Improved by Electroacupuncture in Patients with Mild Cognitive Impairment? A Pilot Randomized Controlled Trial” proposed an investigation on the impact of electroacupuncture (EA) intervention on five cognitive functions (i.e., attention, language, visuospatial function, memory, and frontal/executive function) in persons with a diagnosis of mild cognitive impairment (MCI). Sixty persons with MCI were randomly assigned to one of the following groups: 1) EA intervention, 2) sham EA, and 3) usual care. Intervention consisted of 24 sessions two times per week. A standardized neuropsychological battery was administered at baseline, post-test, and follow-up after 12 weeks from the end of the intervention. Forty-five participants completed the study. Results showed that scores of visuospatial function tended to be higher in the EA group with respect to usual care group at post-test evaluation. Authors discussed their results in light of previous literature as well as the limitations of their study, and gave hints for future research.

I carefully read the manuscript, and I think it may be of interest for the readers of Brain Sciences. The manuscript is very well-written and properly addresses the interesting issue of the effect of electroacupuncture on cognitive functions in the preclinical stages of dementias. It is also relevant that the Authors used two groups (sham EA, usual care) in order to compare the results obtained by the EA group.

I found that the manuscript is really well-written. The introduction section as well as the aims of the study are clear and detailed. The methodology is very rigorous and accurate, as well as the explanations provided in the discussion section. Below there are my comments and suggestions.

Introduction section

Page 1, line 38: maybe Authors intended “functions” instead of “decline”.

Page 2 lines 73-77: Authors often refer to some studies without citing them. Please, can you cite specific studies for each statement that you introduce?

Materials and Methods section

Page 3 lines 97-98: there seems to be a discrepancy among the date of the first participant recruitment presented in line 94 (October 31, 2016) and the date presented in lines 97-98 (November 10, 2016).

Page 4-5 lines 175-176: Authors wrote “Fourteen non-classical acupoints located in the upper limbs and lower limbs were selected.” Why did you choose different and non-classical acupoints for sham EA condition with respect to those chosen for EA condition?

Page 5 lines 217-220: Please, be more specific about the description of the PGIC. What verbal formula was used? Did you ask about the whole cognitive functioning or about single cognitive domains?

Results section

Page 14 Table 6: Values reported in the last line of the table (“New Blind Index”) are different with respect to those reported in line 375. Indeed, they seems to be a copy-and-paste of the values for the blind index of the week 0.

Author Response

Thank you for giving us lots of valuable comments and the opportunity to improve our manuscript. Please see the attachment for our point-by-point responses.

Comment 1:

Introduction section

Page 1, line 38: maybe Authors intended “functions” instead of “decline”.

Response 1:

Thank you for your specific comments to improve our manuscript. We changed the word as the suggestion.

(Page 1, line 40) Through neuroprotective mechanisms, acupuncture-related treatment has potential to improve cognitive functions in animal studies

Comment 2:

Introduction section

Page 2 lines 73-77: Authors often refer to some studies without citing them. Please, can you cite specific studies for each statement that you introduce?

Response 2:

Thank you for your valuable comments. Related specific references have been added.

(Page 2, lines 76-79) In addition, most previous studies have mainly used active controls such as donepezil and nimodipine [4,5,19]. Few studies have used a sham control [21,22]. As a result, there has been limited research about the specific effect of EA on cognitive impairment. Furthermore, most studies have a treatment period of 4-8 weeks [4,5,20]. Few studies have a long-term follow-up of more than 24 weeks.

[4, 5]

Kim, H., et al., Cognitive improvement effects of electro-acupuncture for the treatment of MCI compared with Western medications: a systematic review and Meta-analysis. BMC complementary and alternative medicine, 2019. 19(1): p. 13.

Deng, M. and X.F. Wang, Acupuncture for amnestic mild cognitive impairment: a meta-analysis of randomised controlled trials. Acupunct Med, 2016. 34(5): p. 342-348.

[19]

Zhang, H., et al., Clinical observation on effect of scalp electroacupuncture for mild cognitive impairment. Journal of Traditional Chinese Medicine, 2013. 33(1): p. 46-50.

[21, 22]

Jia, B., et al., The effects of acupuncture at real or sham acupoints on the intrinsic brain activity in mild cognitive impairment patients. Evidence-Based Complementary and Alternative Medicine, 2015. 2015.

Tan, T.-t., et al., Modulatory effects of acupuncture on brain networks in mild cognitive impairment patients. Neural regeneration research, 2017. 12(2): p. 250.

[20]

Kim, J.-H., et al., Cognitive Improvement Effects of Electroacupuncture Combined with Computer-Based Cognitive Rehabilitation in Patients with Mild Cognitive Impairment: A Randomized Controlled Trial. Brain sciences, 2020. 10(12): p. 984.

Comment 3:

Materials and Methods section

Page 3 lines 97-98: there seems to be a discrepancy among the date of the first participant recruitment presented in line 94 (October 31, 2016) and the date presented in lines 97-98 (November 10, 2016).

Response 3:

Thank you for your valuable comments. The screening date of the first participants was October 31, 2016 was the screening date of the first participants, and November 10, 2016 was the enrollment date of the first participants. We revised the date of recruitment period based on the screening date as below.

(Page 3, line 99-100) Sixty eligible participants in two Korean medicine university hospitals from Octorber 31, 2016 to August 24, 2018 were recruited.

Comment 4:

Materials and Methods section

Page 4-5 lines 175-176: Authors wrote “Fourteen non-classical acupoints located in the upper limbs and lower limbs were selected.” Why did you choose different and non-classical acupoints for sham EA condition with respect to those chosen for EA condition?

Response 4:

Thank you for your valuable comments. According to the review report about the sham electroacupuncture methods in randomized controlled trials (Chen, 2017), sham EA method can be examined according to three respects: needle location, depth of needle insertion, and electrical stimulation. They summarized seventeen kinds of sham EA methods, and we used the method of ‘Sham EA on non-specific acupuncture points + no skin penetration + no electric stimulation” Acupoints for EA condition were selected based on previous research of EA on cognitive disorders. Non-classical acupoints for sham EA condition were selected to avoid specific points which are known to be related to improve cognitive functions. We revised the method as below.

(Page 5, lines 178-180) Fourteen non-classical acupoints located in the upper limbs and lower limbs were selected to avoid the location-specific effect of EA. We chose sham acupoint locations not sharing the dermatomes with acupoints used in the EA group [31].

Chen Z, Li Y, Zhang X, et al. Sham Electroacupuncture Methods in Randomized Controlled Trials. Scientific Reports. 2017;7(1):1-19. doi:10.1038/srep40837

Comment 5:

Materials and Methods section

Page 5 lines 217-220: Please, be more specific about the description of the PGIC. What verbal formula was used? Did you ask about the whole cognitive functioning or about single cognitive domains?

Response 5:

Thank you for your valuable comments. We asked participants about the whole cognitive functioning. More specific information including verbal formula used for PGIC were added in the method section.

(Page 5 lines 221-224) Additionally, Patient Global Impression of Change (PGIC) was examined at the post-treatment and follow-up assessment. The following question was asked to the participants. “How much did cognitive impairment improve compared to before participation in the study?” Participants were asked to choose answers among “very much improved”, “much improved”, “minimally improved”, “no change”, “minimally worse”, “much worse”, and “very much worse”.

Comment 6:

Results section

Page 14 Table 6: Values reported in the last line of the table (“New Blind Index”) are different with respect to those reported in line 375. Indeed, they seems to be a copy-and-paste of the values for the blind index of the week 0.

Response 6:

Thank you for your valuable comments. As you mentioned, there was an error in the value. Correction has been made. Thank you.

(Page 14, Table 6)

Round 2

Reviewer 1 Report

These changes improved the article's title and content.